# What are the barriers and facilitators to self-management of chronic conditions reported by women? A systematic review

Lucy Dwyer  ,[1,2] Dawn Dowding,[2] Rohna Kearney[1,3]

[1]The Warrell Unit, Manchester University NHS Foundation Trust, Manchester Academic Health Science Centre, Manchester, UK
[2]Division of Nursing, Midwifery and Social Work, The University of Manchester School of Health Sciences, Manchester, UK
[3]Institute of Human Development, Faculty of Medical & Human Sciences, University of Manchester, Manchester, UK

**Correspondence to**
Lucy Dwyer;
lucy.dwyer@postgrad.manchester.ac.uk

## ABSTRACT

**Introduction** Pelvic organ prolapse (POP) can be effectively managed using a pessary. A scoping review found that pessary self-management appears to benefit women with no increased risk. Despite this, many are unwilling to self-manage their pessary. At present, there is a lack of understanding about what affects willingness to self-manage a pessary. However, there may be relevant, transferable findings from other literature about barriers to the self-management of other chronic conditions. Therefore, this systematic review aims to identify, appraise and synthesise the findings of published qualitative research exploring the barriers and facilitators to self-management of chronic conditions reported by women.

**Methods and analysis** The systematic review will be conducted and reported in accordance with Preferred Reporting Items for Systematic Reviews and Meta-Analyses (PRISMA) guidelines and a guide for the systematic review of qualitative data. A search of MEDLINE, CINAHL, Embase and PsycInfo will be undertaken to identify relevant articles that meet the eligibility criteria using the search terms 'Women', 'Woman' 'Female,' 'Chronic', 'Long-term', 'Disease', 'Illness', 'Condition' 'Health,' 'Self-management,' 'Qualitative,' 'Barrier' and 'Facilitator'. A hand search of the reference list of non-original research identified during the search but excluded will be conducted for additional publications, which meet the inclusion and exclusion criteria. Studies published before 2005 and those not available in English will be excluded. Data relevant to the topic will be extracted and critical appraisal of all included publications undertaken.

**Ethics and dissemination** No ethical or Health Research Authority approval is required to undertake the systematic review. The systematic review findings will be disseminated by publication. The findings will also inform subsequent exploratory work regarding pessary self-management.

**PROSPERO registration number** CRD42022327643.

## STRENGTHS AND LIMITATIONS OF THIS STUDY

⇒ The review will be conducted using rigorous and reproducible methods to systematically identify, appraise and synthesise relevant qualitative evidence.
⇒ Critical appraisal of the evidence will be undertaken to determine the weighting of included research findings, as well as identifying the methodological strengths and limitations of the current evidence base.
⇒ The identification and synthesis of data will be limited to published articles found on the MEDLINE, CINAHL, Embase and PsycInfo databases and a hand search of reference lists. The authors will not seek original data.
⇒ The criteria that included studies must be published in English means there is potential for cultural bias in the findings.

## INTRODUCTION

Pelvic organ prolapse (POP) is the downward displacement of one or more of the pelvic organs including the uterus, vaginal compartments, bowel or bladder.[1] Symptoms of POP include a vaginal bulge, heaviness or a dragging sensation, difficulties voiding or defecating and sexual dysfunction, all of which can significantly negatively impact a woman's quality of life.[1] A pessary is a medical device that can be inserted into the vagina to provide mechanical support to the prolapsed organs.[2] Pessaries offer women with prolapse a comparable improvement to surgery.[3 4] However, the need for regular follow-up deters some women from this treatment option.[5–7] It has been suggested that women could be supported to self-manage their pessary, removing and inserting it independently, which would facilitate increased autonomy and potentially reduce the amount of face to face appointments required.[8] A scoping review conducted by the authors suggested pessary self-management appears to offer benefits such as comfort, convenience, increased perception of help and support, and autonomy with no increased risk of complications (Dwyer *et al*, 2022, in press). However, the review also found that many women do not feel able, or lack willingness to self-manage their pessary, concluding

that further research was required to explore barriers to self-management (Dwyer *et al*, 2022 in press).

## Rationale

There is currently a lack of evidence about barriers to pessary self-management; however, there may be relevant, transferable findings from other literature about barriers to the self-management of other chronic conditions. POP is a condition that solely affects women. Therefore, further understanding of the barriers to self-management specifically in women is indicated to ensure findings are relevant and generalisable to the female population. It is acknowledged that the intimate nature of pessary self-insertion and removal may contribute to condition-specific barriers as identified in the scoping review (Dwyer *et al*, 2022, in press). However, it is also possible there are barriers to pessary self-management that apply to the concept of self-management of any chronic condition. The term self-management was first used by Creer during the 1960s to describe increased patient participation and engagement with care planning and treatment.[9] This definition of self-management has continued to evolve, and the current definition describes self-management as a procedure where a patient changes their behaviour through goal setting, information utilisation, decision making, action and self-reaction with the aim of improving health, quality of life and increased self-efficacy.[10] The terms self-management and self-care have been used interchangeably and without clear definition, which has led to confusion between both behaviours.[9] The WHO[11] and UK Department of Health[12] define self-care as actions to maintain and improve overall health as part of daily living.[9] Whereas self-management is condition focused.[9]

## Objectives

This systematic review aims to identify, appraise and synthesise the findings of published qualitative research exploring the barriers and facilitators to self-management of chronic conditions reported by women. These findings will guide further research exploring and intervention development to support women to overcome barriers to pessary self-management.

## METHODS AND ANALYSIS

Qualitative research is the most appropriate evidence to explore the barriers and facilitators to self-management as it offers insight into the experiences and perceptions of individuals.[13] There are extensive tools and literature to guide the systematic review process of quantitative data; however, there are fewer resources available to ensure a robust systematic review of qualitative data.[14] Therefore, in addition to using Preferred Reporting Items for Systematic Reviews and Meta-Analyses (PRISMA) guidance (online supplemental file 1),[15] a guide for the systematic review of qualitative data has also been used.[14] This will ensure the systematic review process meets the

high-quality standards established by PRISMA but also takes into account the methodological differences of qualitative and quantitative research.

In view of the focus on qualitative publications, instead of the Participants, Intervention, Comparators and Outcomes tool advocated by PRISMA,[15] a modified tool will be used. Because qualitative studies do not have interventions and comparators, defining the relevant Population, Context and Outcome (PCO) is more appropriate.[14] Using the PCO tool, the population was specified as women aged 18 years or older, the context is studies exploring interventions designed to support the self-management of chronic conditions. The outcome is qualitative data reported by women that is relevant to barriers and facilitators that influence the uptake, maintenance or concordance of the self-management intervention. Therefore, using the previous parameters using the PCO tool, the following research question was formulated: what are the barriers and facilitators to self-management of a chronic condition reported by women?

## Registration

In accordance with PRISMA recommendations, this systematic review has been registered with The Open Science Framework (10.17605/OSF.IO/CTHSF) and PROSPERO (CRD42022327643) to ensure transparency and prevent contemporaneous duplication of the review.[14 15]

## Information sources

The systematic qualitative review will be conducted in accordance with PRISMA guidelines[15] (online supplemental file 1). A systematic search of Medline, CINAHL, PsychINFO and Embase databases will be carried out using the search terms detailed in table 1 (online supplemental file 2). The databases were identified following a discussion with an information technician at The University of Manchester library based on their coverage of medical, nursing and allied health, psychology and biomedical journals.

## Eligibility criteria

Using the criteria detailed in table 2, the reviewer will assess each identified publication's eligibility for inclusion. A PRISMA flow diagram will be used to present the screening process including the number of studies excluded and reasons for this to ensure transparency and reproducibility.[15]

## Data management and selection

Following the search, all identified abstracts will be uploaded to reference manager software, following which duplicates and non-original research publications will be removed. The abstracts will then be reviewed for relevance to the review question and in accordance with the eligibility criteria. A sample of 20% of abstracts will be screened by an independent reviewer to ensure concordance with decisions about eligibility. In the instance of disagreement regarding included or excluded studies not

**Table 1** Search terms

| Population | Context self-management | Context chronic condition | Context qualitative methodology | Outcome |
|---|---|---|---|---|
| Women | Self-management | Chronic | Qualitative | Barrier* |
| Woman | | Long term | | Facilitator* |
| Female | | Disease* | | |
| | | Illness* | | |
| | | Condition* | | |
| | | Health | | |

*Search terms to identify both singular and plural variations of the word, for example, barrier and barriers.[24]

resolved through discussion, a third reviewer will be asked to make the final decision. Potentially relevant sources will be retrieved in full and assessed in detail against the inclusion criteria. Reasons for exclusion of sources of evidence that do not meet the inclusion criteria will be recorded and reported in the systematic review. The reference list of all review articles identified in the search and excluded due to not being original research will be checked, and all relevant original research cited will be reviewed for potential inclusion within this review.

### Data collection

Data will be extracted using a data extraction tool developed by the reviewers based on qualitative systematic review guidance.[14 16] The extracted data will be entered into the tool electronically via Microsoft Excel. The data extraction form will include details of the author(s), year of publication, methodology utilised, theoretical orientation of the researchers, population studied, the self-management intervention, the chronic condition targeted and key findings relevant to the barriers and facilitators of the self-management intervention. Study findings will be recorded in two separate columns to differentiate between first and second order constructs maintaining clarity about what findings are raw data and which are researcher interpretations.[14]

A second reviewer will perform data extraction from a subset of 10% of included articles to ensure a standardised, reproducible approach. In the instance of discrepancies in data extraction processes, these will be explored, discussed and changes made to the data extraction tool if necessary. Any amendments to the data extraction tool will be recorded and published with the review findings to ensure transparency. The final version of the data extraction tool including extracted data will be published alongside the review findings to ensure reproducibility.

### Data synthesis

The extracted data will be analysed using a thematic approach. Findings will be coded line by line to identify emerging and recurring descriptive themes among the studies.[17] Once all extracted data have been coded, the descriptive themes identified from the coding will be further analysed for broader analytic themes that span the included studies.[17] A number of qualitative systematic reviews of barriers and facilitators to healthy behaviour have used thematic synthesis.[18] Therefore, the synthesis will be performed using this established process to ensure an established inductive and deductive approach.[18]

### Quality appraisal

Qualitative findings offers insight into individuals' views, experiences and perspectives and therefore do not attempt to be generalisable or replicable.[19] However, rigour can be achieved in qualitative research by ensuring findings are trustworthy, authentic, typical, transferable and valid.[19] Butler et al[14] describe their use of the Critical Appraisal Skills Programme (CASP) qualitative checklist.[20] CASP specify they have not suggested a scoring system for appraising qualitative studies as this was not the purpose of the checklist.[20] However, Butler et al[14] explain the scoring system they successfully used for each checklist criterion and overall scoring to determine the level of quality of each individual study. Therefore, this same scoring system of the CASP checklist will be used (table 3). As this review aims to provide an overview of the current evidence base, all studies will be included regardless of the appraised quality.[14] However, as advocated by Long et al,[21] a hierarchical approach to thematic analysis will be undertaken with the quality of a study informing whether or not new codes are created. For this review, new codes will be created for high and medium quality studies. However, for low and very low-quality studies, no new codes will be created but findings used to support themes identified from codes created through thematic analysis of higher quality studies. This will ensure finding are weighted appropriately based on the methodological quality of the studies.[21] Furthermore, an overview of strengths and limitations of the identified evidence base will be provided to highlight any patterns in quality, or lack of it, and inform the methodological design of future research within this subject area.

Another issue with the synthesis of findings from qualitative studies is that the subjectivity of study findings means our confidence in cumulative evidence synthesised from included studies is reliant on decisions and interpretations

**Table 2** Eligibility criteria

| Criteria | Explanation | Justification |
|---|---|---|
| Studies where participants are women | Studies with data from both men and women can be included if the findings are differentiated between men and women's responses. | POP is a condition that solely affects women. Therefore, to ensure the findings of this review can be generalised to the relevant population, it is necessary to limit included studies in case there are differences between the barriers and facilitators men and women report and experience to self-management interventions. |
| Studies where participants are aged 18 years or over | Studies with data from individuals over and under 18 years can be included if the findings are differentiated between age groups. | The incidence of POP among women aged under 18 years is extremely low.[25] Therefore, the focus of research exploring pessary self-management will be women aged over 18 years. To ensure the findings of this review can be generalised to the relevant population, it is necessary to limit included studies in case there are differences between the barriers and facilitators adult women and young women and girls report and experience to self-management interventions. |
| Studies exploring a self-management intervention | An intervention designed with the aim of improving health, quality of life or perceived self-efficacy for individuals with a specific condition through goal setting, information utilisation, decision making, action and self-reaction.[10] | The terms self-care and self-management have been poorly defined and used interchangeably.[9] Therefore, it is necessary to ensure that self-management interventions included within the review meet Creer's definition[10] that a self-management intervention aims to improve health, quality of life or perceived self-efficacy for an individual with a specific health condition through either goal setting, information provision, supported decision making or self-reaction, to ensure the findings are valid. |
| Studies exploring a self-management intervention for a chronic condition | The term chronic disease has been poorly defined within both guidelines and literature leading to confusion.[26] Therefore, to ensure a standardised approach for the purpose of this review, a chronic (also known as a long-term) condition will be defined as a non-communicable condition that lasts for 6 months or longer and impacts on quality of life.[26–28] | POP meets this definition of a chronic condition, therefore to ensure the findings of other studies included within the review are likely to be transferable to women with POP, it is necessary to exclude interventions for acute or terminal conditions as the barriers and facilitators cited may differ significantly to individuals with chronic conditions. |
| Studies that are qualitative research | Original qualitative research. Qualitative studies, mixed methods studies and data collected by questionnaire can be included if answered in free text and analysed using a qualitative approach. | Qualitative research is the most appropriate evidence to explore the barriers and facilitators to self-management as it offers insight into the experiences and perceptions of individuals.[13] |
| Studies which have findings related to the barriers or facilitators to a self-management intervention | The results will include findings about the barriers and facilitators which influence uptake and/or maintenance and /or concordance with a self-management intervention. | The aim of the review is to identify the barriers and facilitators to self-management interventions. |
| Studies with finding directly reported by the target population of women | Data about barriers and facilitators to a self-management intervention must be collected directly from the study population rather than healthcare professionals, partners, family members or carers. | This systematic review aims to identify barriers and facilitators to self-management interventions from the perspective of the women they are designed for. Therefore, the perspectives of healthcare professionals, partners, family members or carers will not be included as these interpretations may not accurately reflect the experiences of women. |
| Language | Studies must be published in the English language. | For pragmatic reasons, studies published in languages other than English will be excluded from the review due to the cost and time required for translation. |
| Original research | Publications that are not original research including reviews, case reports and commentary articles will not be included. | Non-original research will be excluded to avoid duplication of results and ensure all included publications provide rigorous evidence rather than subjective opinion. |

Continued

**Table 2** Continued

| Criteria | Explanation | Justification |
|---|---|---|
| Date range | Eligible studies will be published between 2005 and the date of the search. | Since 2005, UK key health policy demonstrates a distinct change from a traditional medical model approach to healthcare to a person-centred approach where individuals are empowered to self-care and self-manage their long-term conditions with the goal of providing a more efficient service.[12 29 30] The NHS Long-term Plan, launched in 2019, refers to supported self-management as a means to meet healthcare delivery goals and improve patient care,[31] demonstrating that self-management remains a key aspect of national healthcare policy. Therefore, the time parameters for the search will be from 2005 until the search date. It is anticipated this will ensure review findings are sufficiently current to reflect recent practice and experience of women. |

NHS, National Health Service; POP, pelvic organ prolapse.

made by authors. While quality appraisal using the CASP tool assesses research processes, it does not determine the credibility or dependability of interpretations made by the researcher that may be influenced by personal bias.[22] Because of this, researchers from the Joanna Briggs Institute developed a system called ConQual, which enables assessment and scoring of a study's credibility, dependability and therefore overall confidence in the synthesised findings.[22] Using the ConQual, statements made from the findings of synthesised data will be presented in tabular format alongside a ConQual rating of credibility, dependability and overall score to ensure conclusions made can be weighted accordingly.

### Outcomes and prioritisation

The experiences and perceptions of women reporting barriers and facilitators to self-management interventions targeting chronic diseases are the focus of this review. Therefore, all barriers or facilitators to the uptake or maintenance of a self-management intervention for a chronic condition reported within identified and eligible studies will be included within data synthesis.

**Table 3** Quality appraisal scoring system

| CASP scoring system | |
|---|---|
| **Checklist criterion** | |
| Yes | 1 |
| No | 0 |
| Unsure | 0.5 |
| **Overall score** | |
| High quality | 9–10 |
| Moderate quality | 7.5–8.5 |
| Low quality | 6–≤7.5 |
| Very low quality | ≤6 |

CASP, Critical Appraisal Skills Programme.

### Data analysis and presentation

Details of the studies identified by the search, for example, the extent of identified literature, the context of included research such as the methodology used, philosophical standpoint, country of origin, study population, self-management intervention and chronic disease targeted, will be presented in numerical and tabular format to provide an overview of the evidence base. A CASP and ConQual ratings will also be presented alongside each included study to enable readers to take into account the quality, dependability, credibility and overall confidence in each study and therefore synthesised findings. Themes identified will be described in both tabular and text format to ensure transparency and clarity in the process of thematic analysis and identified themes, in addition to more detailed discussion and demonstration of evidence supporting identified themes within the text.

### Confidence in cumulative evidence

One criticism of the synthesis of qualitative studies is that the nature of qualitative research means the context of each study may vary significantly, impacting on study findings that creates a potential for bias when performing meta-synthesis.[17] To minimise this, the extraction tool will include details of each study's context and be presented alongside the review findings, enabling readers to contextualise study findings as advocated by Thomas and Harden.[17] Further subanalysis will be undertaken to explore whether certain contexts, for example, the age of a study population, type of self-management intervention or chronic condition correlate with specific themes. This will further understanding about whether identified barriers or facilitators relate to a specific context or have broader applicability.[17]

### Patient and public involvement

Members of the public have not directly been involved with development of this protocol or review process. However, the need for research exploring pessary

self-management was highlighted by The James Lind Alliance (JLA) Priority Setting Partnership for pessary and prolapse.[23] Several women with experience of pessaries participated in this partnership either as members of the steering group, by attending the consensus workshop or completing questionnaires. Understanding more about pessary self-management was ranked third out of 20 priorities by the JLA Priority Setting Partnership. This systematic review will inform a research project exploring pessary self-management, which has been identified and prioritised by patients and members of the public.

## Ethics and dissemination

No ethical or Health Research Authority approval is required to undertake the systematic review. The systematic review findings will be disseminated by publication in a peer-reviewed journal. The findings will also inform subsequent exploratory work regarding pessary self-management.

**Contributors** LD devised the systematic review question, methodology and drafted this manuscript. DD substantively contributed to the development of the systematic review question, methodology and revised and approved this manuscript. RK substantively contributed to the development of the systematic review question, methodology and revised and approved this manuscript.

**Funding** LD, Clinical Doctoral Research Fellow, NIHR300519, is funded by Health Education England/National Institute for Health Research (NIHR) for this research project.

**Disclaimer** The views expressed in this publication are those of the author(s) and not necessarily those of the NIHR, NHS or the UK Department of Health and Social Care.

**Competing interests** LD and RK are coapplicants of the NIHR/HTA funded Treatment of Prolapse with Self-Care Pessary study.

**Patient and public involvement** Patients and/or the public were not involved in the design, or conduct, or reporting, or dissemination plans of this research.

**Patient consent for publication** Not applicable.

**Provenance and peer review** Not commissioned; externally peer reviewed.

**ORCID iD**
Lucy Dwyer http://orcid.org/0000-0002-0284-873X

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
