## [Reviewer comments · BMJ Open]

ARTICLE DETAILS

TITLE (PROVISIONAL)	What are the barriers and facilitators to self-management of chronic conditions reported by women? A systematic review.
AUTHORS	Dwyer, Lucy; Dowding, Dawn; Kearney, R

VERSION 1 – REVIEW

REVIEWER	Rantell, Angela King's College Hospital, Department of Urogynaecology
REVIEW RETURNED	22-Feb-2022

GENERAL COMMENTS	Thank you very much. This was a very comprehensive article and protocol and i do not recommend any changes.
---

REVIEWER	Brown, Claire Cambridge University, Physiotherapy
REVIEW RETURNED	25-Feb-2022

GENERAL COMMENTS	there is a typo in line 38 - double comma
---

REVIEWER	Tooth, Leigh The University of Queensland, School of Public Health
REVIEW RETURNED	28-Mar-2022

GENERAL COMMENTS	protocol - What are the barriers and facilitators to self-management of chronic conditions reported by women? A systematic review. This protocol is well written. I only have a few comments. I note there are no protocol limitations mentioned. Please include earlier (in abstract, in the text on page 10) that the review will include articles from 2005 to present and only papers in English. On page 8 please specify which Country's Department of Health you refer to. On page 9 why are not also registering this with PROSPERO? It is my understanding that they accept SRs of qualitative interventions.
--

VERSION 1 – AUTHOR RESPONSE

Please include details of your registration in OSF at the end of your Abstract under the heading "Registration".	Actioned
- Along with your revised manuscript, please include a copy of the PRISMA-P checklist indicating the page/line numbers of your manuscript where the relevant information can be found	This was submitted as supplementary material, please can you confirm whether there is an issue with the document submitted?
Please include, as a supplementary file, the precise, full search strategy (or strategies) for all databases, registers and websites, including any filters and limits used.	Uploaded as supplementary file
Please remove all your figures in your main document and upload each of them separately under file designation 'Image' (except tables and please ensure that figures are in better quality or not pixelated when zoomed in). They can be in TIFF, JPG or PDF format. Make sure that they have a resolution of at least 300 dpi and at least 90mm x 90mm of width. Figures in document, excel and powerpoint format are not acceptable.	Resolution has been increased to 330 dpi. All figures included are tables therefore I am uncertain from the comment about what action to take. Is it acceptable to leave these in the main document and in word format?
There is a typo in line 38 - double comma	Actioned
I note there are no protocol limitations mentioned.	Apologies, I am not certain whether or not I have correctly understood this comment. I have interpreted it as limitations on the search strategy which I have rectified as detailed below.
Please include earlier (in abstract, in the text on page 10) that the review will include articles from 2005 to present and only papers in English.	As requested I have added this to the abstract. These aspects of the exclusion criteria are also referred to under the inclusion/exclusion criteria subheading of the methods section.
On page 8 please specify which Country's Department of Health you refer to.	Actioned
On page 9 why are not also registering this with PROSPERO? It is my understanding that they accept SRs of qualitative interventions.	Based upon your recommendation, we have also registered this review with PROSPERO.